

# The influence of global climate and local hydrological variations over streamflow extremes: The tropical-mountain case

Juan Contreras[1], Daniel Mendoza[2], Jheimy Pacheco[3], Alex Avilés[1]

[1]Carrera de Ingeniería Ambiental, Facultad de Ciencias Químicas, Universidad de Cuenca, Av. 12 de abril s/n y Loja, Cuenca, 010203, Ecuador

[2]Carrera de Ingeniería Civil & Departamento de Ingeniería Civil, Facultad de Ingeniería, Universidad de Cuenca, Av. 12 de abril s/n y Loja, Cuenca, 010203, Ecuador

[3]Instituto de Estudios de Régimen Seccional del Ecuador (IERSE), Universidad del Azuay, Av. 24 de mayo 7-77 y Hernán Malo, Cuenca, 010204, Ecuador

*Correspondence to*: Alex Avilés (alex.aviles@ucuenca.edu.ec)

**Abstract.** Hydrological extremes such as floods and droughts are the most common and threatening natural disasters worldwide. Particularly, tropical Andean headwaters systems are prone to hazards due to their complex climate conditions. However, little is known about the underlying mechanisms triggering such extremes events. In this study, the Generalized Additive Models for Location, Scale and Shape (GAMLSS) were used for investigating the relations between the Annual-Peak-Flows (APF) and Annual-Low-Flows (ALF), respecting to climate and land use/land cover (LULC) changes. Thirty years of daily streamflow data-sets taken from two Andean catchments of southern Ecuador are used for the experimental research. Global climate indices (CI), describing the large-scale climate variability were used as hypothetical drivers explaining the extreme's variations on streamflow measures. Additionally, the Antecedent-Cumulative-Precipitation (AP) and the Standardized-Precipitation-Index (SPI), and LULC percentages were also included as possible direct drivers – synthetizing local climate conditions and localized hydrological changes. The results indicate that AP and SPI clearly explain the extreme streamflow variability. Nonetheless, global variables play a significant role underneath the local climate. For instance, ENSO and CAR exert influence over the APF, while ENSO, TSA, PDO and AMO control ALF. Furthermore, it was found that LULC changes strongly influence both extremes; although this is particularly important for relative more disturbed catchments. These results provide valuable insights for future forecasting of floods and droughts based on precipitation and climate indices, and for the development of mitigation strategies for mountain catchments.

## 1 Introduction

Hydrological extremes such as floods and droughts are the most common and threatening natural disasters worldwide. Frequency analyses of extreme events have been widely based on the assumption of stationarity (Serago and Vogel, 2018). However, recent studies have shown that hydrometeorological extreme events have changed under the influences of climate change and human activities (Jongman et al., 2014; Min et al., 2011). This fact is the result of the absence of stationarity of the hydrometeorological variables (Milly et al., 2008). Under environmental changing conditions, streamflow (ranging from low to high) could be affected because of alterations on any of its environmental drivers, such as precipitation, temperature



and land use and land cover (LULC). Generally, small changes of these drivers entail a significant impact on frequencies and
magnitudes of floods; also, they modify the characteristics of low-extremes (Xiong et al., 2018). Therefore, the identification
of such changes and their underlying drivers is essential to perform an adequate analysis of hydrological extreme series, as
well as it is for decision-making regarding the integrated water resources management.

Previous research treating the non-stationarity features of hydrological series was mainly focused on two aspects: (i) the
development of methods dealing with the non-stationary properties; and (ii) the identification of external variables
(covariates) handling these non-stationary features. This latter has attracted increased attention last years (e.g. Neri et al.
2019; Slater and Villarini 2017; Xiong et al. 2018) and has been identified as one of the key research topics by the
International Association of Hydrological Sciences (IAHS; Blöschl et al. 2019) for coming years. The Generalized Additive
Models for Location, Scale and Shape (GAMLSS) has attracted attention because of its great flexibility and good
performance (Slater et al., 2019; Villarini et al., 2009; Yan et al., 2017a). This framework firstly proposed by Rigby and
Stasinopoulos (2005), is able to deal with non-stationary properties of the series, following principles under the time-series
analysis framework. A common approach for handling such non-stationary behavior is to hypothesize alternative covariates
driving the time variation features of a particular data-set.

Several studies have taken into account different physical covariates in the non-stationary analysis of hydrological extremes
to reflect the impact of climate and human activities. On one hand, climate variability has been commonly represented by
large scale climate indices (Gu et al., 2016; Ishak et al., 2013; Li et al., 2015; Li and Tan, 2015; López and Francés, 2013;
Villarini et al., 2012; Zhang et al., 2015) and by annual/seasonal local precipitation and temperature (Du et al., 2015; Neri et
al., 2019; Slater and Villarini, 2017; Villarini and Strong, 2014; Xiong et al., 2018; Yan et al., 2017a, 2017b). On the other
hand, dynamics of human activities have been included using population density (Neri et al., 2019; Slater and Villarini,
2017; Villarini et al., 2009), coverage of agricultural (Neri et al., 2019; Slater and Villarini, 2017; Villarini and Strong, 2014)
and urbanized areas (Prosdocimi et al., 2015), as well as multiple reservoir indices (Li and Tan, 2015; López and Francés,
2013; Su and Chen, 2019a; Zhang et al., 2015).

The tropical Andean region is considered as an area of high vulnerability to climate change and hydroclimatic risks (Field et
al., 2012). Mainly, Andean headwater catchments which are known to provide multiple ecohydrological services (Aparecido
et al., 2018; Célleri and Feyen, 2009; Lazo et al., 2019) are prone to hazards due to their complex terrain and their high
spatio-temporal climate variability (Muñoz et al., 2018). On the one hand, studies have reported an increase in temperature
and in annual precipitation in most sites of the Andean region of Ecuador (Celleri et al., 2007; Morán-Tejeda et al., 2016).
Though, precipitation has become more extreme in rainy periods showing an increase for wet periods (February to April)
and showing a decrease for dry periods (June to September) (Celleri et al., 2007; Tobar and Wyseure, 2018). On the other
hand, land use in the Ecuadorian Andes has experienced complex changes due to different factors such as land reforms,
population growth, migration and increasing agricultural exports in the last decades (Curatola Fernández et al., 2015; Grau
and Aide, 2007; Koning et al., 1999; Vanacker et al., 2003; White and Maldonado, 1991; Wunder, 1996). Several studies
have shown that LULC changes have a great impact on the water availability, regulation, control of erosion, mitigation of

floods and also on diverse ecosystem services in mountain catchments (Bathurst et al., 2011; Bonnesoeur et al., 2019; Buytaert et al., 2006, 2007; Hofstede et al., 2002; Molina et al., 2012, 2015; Ochoa-Tocachi et al., 2016; Tenorio et al., 2018).


Particularly, the Tomebamba catchment is one of the main water suppliers for the city of Cuenca (the third largest city of Ecuador) accounting for almost 34 % of total water demand (Aviles et al., 2016). Currently, studies treating the non-stationarity of hydrological extremes for tropical mountain areas are limited, so they are more focused on large areas (Sedano Cruz, 2017). Sedano Cruz (2017) analyzed the influence of large-scale climate indices and reservoirs on both low

and peak flows in the Cauca river basin. However, local conditions such as precipitation and LULC changes, which could potentially contribute to a better understanding of the hydrological response to patterns of natural and anthropic alteration, have not been investigated yet in small Andean catchments.

In virtue of the last-mentioned, this work focusses its efforts on unveiling the influence that global climate variability and local hydrological alterations exert over the streamflow extremes events. Specifically, research questions we seek to address

were: (1) Which are the large-scale phenomena that control the hydrological extremes in our study area? (2) What are the main drivers (climate or LULC) related of the year-to-year variability in both hydrological extremes? (3) There is a difference in the mechanisms of control of hydrological extremes in a relatively undisturbed and disturbed catchment?

The GAMLSS technique is applied for the purposes set herein. Further, two Andean-mountain sub-catchments systems (belonging to the Tomebamba system) are considered here because their underlying climate and hydrological complex

conditions, which makes of them a natural laboratory for hydrological and climate research.

## 2 Study area and data

### 2.1 Study area

Our study area comprises the drainage area of the Tomebamba catchment at the Matadero en Sayausí station. This area is located in the south-eastern flank of the Andean cordillera in southern Ecuador (Fig. 1). The catchment area is approximately

300 km$^2$ and drains in the direction of the Atlantic Ocean towards the Amazon river. The elevation within the catchment ranges from 2709 m to 4398 m a.s.l. The area corresponds mainly to a páramo ecosystem with wetlands and small lagoons in the highlands. Since 1977, around 40 % of the catchment area belongs to the Cajas National Park (an UNESCO World Biosphere Reserve). The Tomebamba catchment is important for its water supply for domestic, agricultural, industrial and hydropower purposes for the city of Cuenca (over 580,000 inhabitants).

The climate of the study area is driven by continental air masses from the Amazon basin, by the seasonal shift of the Inter Tropical Convergence Zone (ITCZ), and by the cold water upwelling of the Humboldt ocean current (Carrillo-Rojas et al., 2016). The mean daily temperature is ranging from 3 ℃ at the highest peak of the region, to 13.75 ℃ at the outlet of the catchment (Córdova et al., 2016). The annual average precipitation over the studied period is 1049 mm. A bimodal regime is



distinguished in the region with two rainy peaks: one in April and another one in October, with the driest month being
August (Campozano et al., 2016; Celleri et al., 2007).

(Figure 1)

**2.2 Hydrological extreme flow data**

Daily streamflow data from the Surucucho AJ Llullucchas (hereafter Surucucho) and the Matadero en Sayausí (hereafter
Matadero) station (Fig. 1) were obtained through the National Institute of Meteorology and Hydrology of Ecuador
(INAMHI). Daily series covering a period of 30 years (1978-2007) were used to obtain annual peak flow (APF) and annual
low flow (ALF) series. In our study, APF was defined as the annual maximum streamflow selected from daily streamflow
data while ALF was defined as the annual minimum streamflow obtained from monthly streamflow data. Daily streamflow
was used for APF series because floods often occur at short time scales (e.g. hours and days) while monthly streamflow data
are most representative to describe droughts as they occur at large temporal scales (e.g. several days and months). Both
series were calculated for the hydrological year (starting in the driest month), defined from $1^{st}$ of August to $31^{st}$ of July of the
next year for our study area. Details of the streamflow stations and their drainage area are presented in Table 1.

(Table 1)

The APF shows a clear seasonal pattern, with most of the events registered from March to June at both stations. Two peaks,
one in April and another one in June were observed at Surucucho while a single peak in April was observed at Matadero
(Fig. 2a-b). In contrast to APF series, most of ALF were registered from August to February showing a more evenly
distribution in comparison to the APF (Fig. 2). However, a peak is observed in December at Surucucho, and two peaks are
observed at Matadero, one in September and another one during December-January (Fig. 2c-d).

(Figure 2)

**2.3 Covariates**

**2.3.1 Large-scale climate indices (CI)**

Monthly climate indices which potentially modulate precipitation in the study area were downloaded from the National
Oceanic and Atmospheric Administration (NOAA; https://www.esrl.noaa.gov/psd/data/climateindices/list/). Sixteen
candidate climate indices were used: ENSO 1.2, ENSO 3, ENSO 3.4, ENSO 4, Trans Niño Index (TNI), Multivariate ENSO
Index (MEI), Oceanic Niño Index (ONI), Southern Oscillation Index (SOI), Pacific Decadal Oscillations (PDO), Atlantic
Multidecadal Oscillation (AMO), Atlantic Meridional Mode (AMM), North Atlantic Oscillation (NAO), Tropical Northern
Atlantic Index (TNA), Tropical Southern Atlantic Index (TSA), Caribbean Index (CAR) and Quasi-Biennial Oscillation
(QBO). Previous studies have reported that these indices have a significantly influence on precipitation in the study area
(Campozano et al., 2018; Mendoza et al., 2019; Mora and Willems, 2012). To filter out high frequencies that could alter the
relations between the CI and hydrological extremes, a 3-month average of all climate indices prior to the occurrence of the





event-month was calculated. In this way, the CI will exert an influence most likely due to a seasonal persistence inside the natural intra-annual climate cycle. We hypothesized that climate conditions prior to the event-month significantly influence the occurrence of hydrological extremes and are better predictors in comparison to another time window as reported by Zambrano Mera, Rivadeneira Vera, and Pérez-Martín (2018) and Quishpe-Vásquez et al. (2019) for several regions of Ecuador.

### 2.3.2 Precipitation indices (PI)


Daily precipitation was available from 4 rain gauges (Fig. 1), covering the same period of the streamflow data. This information was obtained from the INAMHI and was used to consider local precipitation in the study area. Data gaps totaled less than 5 % for all stations and were filled using the MissForest imputation method (Sidibe et al., 2018; Stekhoven and Bühlmann, 2012). Areal average daily precipitation for the Llaviucu (Surucucho station) and the Tomebamba (Matadero

station) catchments was obtained using the inverse distance weighting (IDW) interpolation method. Two precipitation indices, the cumulative antecedent precipitation and the standardized precipitation index, were used to describe the changes in APF and in ALF, respectively. Details of the construction of these indices are described below.

**Cumulative antecedent precipitation (AP)**

Muñoz et al. (2018) stated that in our study area, flood events are not exclusively caused by extreme precipitation events,

and that non-extreme precipitation events can trigger flash-floods when the soil is saturated due to the high retention capacity of the catchment. Therefore, as a simple way to represent antecedent wetness conditions in the catchment, precipitation of the event-day (AP0) and cumulative antecedent precipitation from 1 till 14 days (i.e., two weeks) prior the occurrence of the event (e.g., AP1, AP2, …AP14) were calculated to be included as explanatory variables. The last antecedent precipitation time-range in the experiment (i.e., AP14), corresponds to the number of days with a considerable probability of consecutive

rainy-days that induced the occurrence of strong floods in the Tomebamba river (Vallejo Llerena, 2014).

**Standardized Precipitation Index (SPI)**

For ALF, the Standardized Precipitation Index (SPI; McKee, Doesken, and Kleist 1993) was included as an explanatory variable. This index has widely been applied for drought characterization (Logan et al., 2010; Vicente-Serrano, 2006; Vicente-Serrano et al., 2012; Vicente-Serrano and López-Moreno, 2005). SPI values were calculated by fitting the Gamma

probability distribution to the monthly interpolated precipitation series at for the Llaviucu and the Tomebamba catchments. The corresponding cumulative probabilities of precipitation were then transformed to the standardized normal distribution where mean was zero and variance one. With the aim of considering the intra-annual variability, SPI at a monthly (SPI1) and at a three-month (SPI3) scale were calculated. In order to consider lag times between changes in precipitation and the occurrence of the ALF, a maximum lead-time of 6 months prior to the event was calculated for both SPI1 (e.g., SPI1_lag0,

SPI1_lag1, …SPI1_lag6) and SPI3 (e.g., SPI3_lag0, SPI3_lag1, …SPI3_lag4) series.



### 2.3.3 Land use and land cover (LULC)

Four LULC maps, dating from 1977, 1990, 2000 and 2008 respectively, at a scale of 1:100000 were obtained, based on national cartography prepared by the Ministry of Environment of Ecuador (MAE; http://mapainteractivo.ambiente.gob.ec/portal). The land use maps were validated and updated through Landsat Thematic
Mapper (TM) and Operational Land Imager (OLI) satellite images, with the exception of the 1977 map, which was obtained from aerial photography. Classification of satellite images was done by means of the maximum likelihood classification technique (Strahler, 1980), which is a supervised classification method. The images were classified and six categories were obtained: forest, lagoons, bare land, páramo grassland, grazing/cultivated land and shrubland (Table 2). The Cellular Automata-Markov model (CA-Markov; Halmy et al. 2015) was used to simulate and predict annual changes in LULC
between 1977 and 2008.

(Table 2)

The CA-Markov is mainly based on three steps: the first step is to obtain a land-use transition probability matrix using Markov chain analysis and applying it on land use maps from 1977 – 1990, 1990 – 2000 and 2000 – 2008. Secondly, neighborhoods have to be identified and finally, the number of iterations which define the best model have to be identified
(Al-sharif and Pradhan, 2014; Yang et al., 2019). The CA-Markov validation was made between the projected map for 2000 and 2008 real map. Kappa indices were used to evaluate the predicted land use map of 2008 with agreement values above 80 % ( $K_{standard}$ = 0.89 and $K_{location}$ = 0.88), indicating very good performance of prediction (Gashaw et al., 2017; Singh et al., 2015). The simulation of change of LULC was made with the IDRISI software.

Annual LULC of forest, páramo grassland (hereafter páramo) and grazing/cultivated land (hereafter grazing) were selected
as candidate explanatory variables as they are the main land uses of the Llaviucu and Tomebamba catchments, accounting for almost 95 % of the entire area and where the biggest LULC changes were observed during the study period (Table 2). The factor used to describe LULC dynamics year to year was the percentage of cover at each catchment.

### 3 Methodology

### 3.1 Detection of trends and change points

With the aim of quantifying trends in the mean of time series, the non-parametric modified Mann-Kendall test was adopted (Hamed and Rao, 1998). The modified Mann-Kendall test is used to remove the effects of autocorrelation found in traditional Mann-Kendall results. A positive value of Zc indicates an increasing trend, a negative value of Zc indicates a decreasing trend, while a zero value of Zc indicates no trend. The null hypothesis of no trend is rejected at the 5 % significance level if |Zc| >1.96, and at the 10 % significance level if |Zc| > 1.645 (Deng et al., 2018; Huang et al., 2013).
Furthermore, to identify change points in the mean of time series, the non-parametric Pettitt test was used (Pettitt, 1979).



## 3.2 Generalized additive models for location, scale, and shape (GAMLSS)

GAMLSS are semi-parametric regression-type models and are widely used to evaluate stationarity and non-stationary of time series (Stasinopoulos et al., 2017). The model allows that all the distribution parameters to be modelled as linear/nonlinear or smooth functions of explanatory variables. GAMLSS has been used to explore the relationships between explanatory variables and response variables and to identify the main influencing factors (Xiong et al., 2018).

GAMLSS assume that a time series $y^T = (y_1, y_2, ..., y_n)$ is a series of independent observations and that it follows a distribution function $F_Y(y_i | \theta_i)$, where $\theta_i = (\theta_1, \theta_2, ... \theta_p)$ is a vector of $p$ distribution parameters representing the location (mean), scale (variance), and shape (e.g., skewness and kurtosis) parameters.

Let $g_k(\bullet)$ $(k = 1, 2, ... p)$ be a known as a monotonic link function linking the distribution parameters to the explanatory variables (covariates). The relationship between the distribution parameters and explanatory variables described by an additive model is:

$$g_k(\theta_k) = X_k \beta_k + \sum_{j=1}^{m} h_{jk}(X_{jk})$$

(1)

where $\theta_k$ and $\beta_k$ are the n-length and m-length parameter vectors, respectively. $X_k$ is fixed known design matrix of explanatory variables of order $n \times m$; while $h_{jk}$ is a smooth non-parametric function of explanatory variables. The regressive relations between the response variable and the covariate can be modeled by linear dependence, represented by the first term on the right hand side of the above equation, or by nonlinear dependence through smoothing terms; represented by the second sum term on the right hand side of the above equation. For the sake of simplicity, we used a linear dependence between the APL/ALF series and covariates, therefore the Eq. (1) can be summarized as:

$$g_k(\theta_k) = X_k \beta_k$$

(2)

Is important to note that linear approximations are the first terms within Tylor's expansions and generally, they represent a large variability proportion into a convergent series. Therefore, linear approximations are reasonable approximations, although the authors acknowledge that any conclusion based on these are not the ultimate word.

## 3.3 Model selection

In order to select an appropriate distribution function for each annual hydrological extreme, a set of distributions widely used for modelling streamflow data was evaluated under stationarity conditions as recommended by Su and Chen (2019). Seven distributions, including six two-parameter distribution functions, were considered in our study: Weibull (WE), Gumbel (GU), Lognormal (LOGNO), Gamma (GA), Logistic (LO), Reverse Gumbel (RG) and one three-parameter distribution function: Generalized Gamma (GGA; or Pearson III distribution). In the case of GGA distribution, considering that the shape





parameter $\theta_3$ is quite sensitive and difficult to estimate, we assumed it to be constant as other studies did (e.g.: Du et al. 2015; López and Francés 2013; Xiong et al. 2018; Yan, Xiong, Guo, et al. 2017). The Akaike Information Criterion (AIC; Akaike 1974) was selected to identify the optimum distribution, which is calculated as:

$$AIC = -2 \, ln \, (ML) + 2k$$

(3)

where $ML$ is the maximum likelihood of the estimated values of the regression parameters and $k$ is the number of parameters within the model. The model with the lowest AIC value is considered the optimal.

Once the best distribution function for each hydrological extreme and station was identified, nine types of models
(considering single and multiple covariates) were developed (Table 3) to evaluate how the structure and the number of explanatory variables influence the performance of the models to describe non-stationary. On the one hand, the models including a single covariate (i.e., M1, M2 and M3) were formed by testing three different combinations as follows: (i) $\theta_1$ parameter is modeled as a function of covariates, but the $\theta_2$ is constant, (ii) the $\theta_1$ parameter is constant, but the $\theta_2$ parameter is modeled as a function of the covariate and (iii) both parameters are modeled as a function of covariates. The
best models are chosen according to the AIC (Eq. 3). On the other hand, to obtain the best models including multiple covariates (i.e., M4, M5, M6, M7 and M8) a forward-stepwise selection process base on the AIC criterion was carried-out for the parameters defining the distributions ($\theta_1$ and $\theta_2$) (Stasinopoulos et al., 2017). The forward approach starts with no explanatory variables in the model, after which it progressively includes the explanatory variables into the model, each time verifying the AIC parameter. In this way, once an optimal subset is chosen for ($\theta_1$), a subsequent optimal set is chosen for
($\theta_2$).

Only significant variables were retained in the final models and were evaluated through the likelihood ratio (LR) test. Thus, nested models were formed to evaluate the significant improvements in the inclusion of additional variables to the models as in other studies (Su and Chen, 2019a, 2019b). Finally, the performances of the non-stationary models were ranked using the AIC (Eq. 3).
245                                                        (Table 3)

### 3.4 Model evaluation

A common practice to evaluate the performance of GAMLSS models is employing diagnostic plots. The centile curves plot and the worm plot were used to support the model selection and to diagnose the fitting performance of the selected models. On the one hand, the centile curves plot presents different percentiles (e.g., 5th, 50th and 95th) to assist the visual inspection
of probabilistic coverage below different percentiles. On another hand, the worm plot is a useful diagnostic tool for the analysis of residuals and is regarded as the detrended Q-Q plot (Yan et al., 2017a). Also, the goodness-of-fit of the models and the independence and normality of the residuals were checked by computing the Filliben coefficient (Filliben, 1975).



Considering our APF/ALF sample size ($n = 30$), a Filliben coefficient higher than 0.963 (p-value < 0.05) indicates a high goodness-of-fit performance of the models.

## 4 Results

### 4.1 Trends and change points of annual hydrological extremes and explanatory variables

Figure 3 shows the APF and ALF series at the Surucucho and Matadero stations. An overall increasing trend was found for both annual hydrological extremes series in both stations during the study period. However, the increasing trends were only significant at the 5 % level (p-value < 0.05) for APF at Matadero and for ALF at Surucucho. Significant (p-value < 0.05) change points were identified for APF series in both stations in 1992 and for ALF series in Surucucho in 1986. A change point for ALF series in the Matadero station was identified in 1981; however, it was not significant. In order to analyze the APF and ALF series before and after the identified change points, the Mann-Kendall and Pettit tests were performed. On the one hand, a decreasing trend is observed before 1992 and an increasing trend after this year for APF at both stations. On another hand, ALF showed a decreasing trend before and after 1986 in Surucucho; while for Matadero, an increasing trend was detected before 1981 and a decreasing trend for the last years. Although these trends were identified, they were no significant. In addition, no change points were detected in these subseries.

(Figure 3)

Table 4 shows the detection of trends and change points of all possible explanatory variables. In general, it is observed that greater changes have occurred in LULC variables in comparison to climate variables (i.e. CI and PI). Significant decreasing trends at the 5% level were observed for forest and páramo land cover while increasing trends were observed for grazing in both catchments. Regarding climate variables, increasing significant trends were observed with CAR and AMO for both stations for all hydrological extreme series. In addition, significant increasing trends were detected in TNA for ALF at both stations, and with ENSO4 and AMM for ALF at Matadero and Surucucho, respectively. Negative trends were observed with TNI for ALF at both stations. In addition, in Matadero, NAO shows a significant negative trend for APF while it is PDO which shows a significant negative trend for ALF. The results of the change point detection showed significant break points for all land cover uses in 1992 at both stations. For climate variables, significant change points in CAR were detected in both stations; in 1996 for APF and in 1993 for ALF, while a significant change point was detected in ENSO4 for ALF at Matadero in 1989. A change point around 1994-1995 was detected in AMO for all hydrological extremes and stations. In addition, significant change points were observed in 1994 for TNA except for APF in Surucucho, while AMM showed a significant break point for ALF at Surucucho in the same year.

(Table 4)



## 4.2 Non-stationary frequency analysis models

As shown in Fig. 4, the best distributions to model APF in Surucucho and in Matadero are GGA and LOGNO, respectively; while the best distributions to model ALF were LOGNO in Surucucho and GA in Matadero. It is important to note that GU

was the worst distribution to model both hydrological extreme series in the stations. With these distributions, single and multiple models were evaluated as mentioned in Sect. 3.3.

(Figure 4)

### 4.2.1 Single covariate models

Table 5 and 6 show the single models and their selected explanatory variables which passed the LR test (p-value < 0.05) for

APF and ALF, respectively. Results showed that different variables are able to explain the variability of hydrological extremes when a single covariate is included. In general, for APF series, models that include PI showed lower AIC values than models which include CI or LULC dynamics; while for ALF, lower values in the AIC were obtained when LULC covariates were used, followed by PI and CI.

For APF, it was found that accumulated precipitation from 1 day to 11 days prior to the event day explains the variability of

APF in Surucucho; while accumulated precipitation of 1 day, 2 days and precipitation from 5 days to 14 days prior the event describe the variability of APF in Matadero. TNI, MEI and ENSO12 were included in the single models in Surucucho and TNI, ENSO12, CAR and ENSO4 were included in Matadero. No LULC uses were suitable to model APF in Surucucho, while for Matadero all LULC uses were suitable to describe the APF; however, LULC variables have slightly lower AIC values than the stationary model.

The lowest AIC values (in order of performance) for each single physical model were M2_AP3 and M1_TNI for Surucucho, while for Matadero M2_AP11, M1_TNI and M3_Grazing were found as lowest values. These results indicate that large scale climate indices such as TNI drives APF in the study area (in both Surucucho and Matadero stations) and that APF is mainly related to cumulative precipitation of 3-days and 11-days before the events in Surucucho and Matadero, respectively. Although grazing showed to be a suitable variable to explain the variability in APF in Matadero, it shows a greater AIC

value than when TNI and AP11 were included.

Regarding to ALF series, the changes in the LULC types used in this study, describe the variability of ALF in Surucucho; however, no LULC variables were included in the Matadero station. In Surucucho, SPI1_lag0, SPI1_lag1, SPI1_lag2, SPI1_lag5, SPI3_lag0, SPI3_lag1 and SPI3_lag2 indices were suitable to explain ALF in this station while SPI1_lag0 and SPI1_lag1 explain the ALF variability in Matadero. AMM, PDO, ENSO4, TNA, AMO, ENSO12 and CAR were related to

the ALF in Surucucho and ENSO3 in Matadero. The best models for each category in Surucucho were M3_Páramo, M2_SPI3_lag2 and M1_AMM, while the lowest AIC values in Matadero were found in M2_SPI1_lag1 and M1_ENSO3.

(Table 5)

(Table 6)





Figure 5 and 6 show the diagnostic assessment of the best single models in both stations to model APF and ALF,
respectively. The majority of the points were within the 5 % and 95 % centile curves, indicating that the models captured the
variability of the data. However, models that include PI are more sophisticated than models that use LULC variables,
following in a better way the dynamics of the annual extreme series. For instance, the best model at Surucucho, which
includes páramo, roughly explain the random temporal variations of ALF. This could be explained because for this particular
case, the overall trend of ALF is better modeled with páramo changes comprising most of the observed points within the 5 %
and 95 % centile curves than climate models although the latter models showed in better way the ALF variability (results not
shown). For both APF and ALF series, all the worm points are within the 95 % confidence intervals, indicating a good fit of
all models. In addition, the Filliben coefficients (Table 5 and 6) of all the models were higher than 0.963, showing good
performance of models.

(Figure 5)

(Figure 6)

### 4.2.2 Multiple covariate models

In order to achieve a more comprehensive picture of the interactions among large scale climate indices and different local
climate and anthropogenic variables, a stepwise procedure was used to incorporate the most influencing variables in the final
models. Only variables that passed the LR test at a significant level of 5 % after the stepwise selection were included in the
multiple covariate models. The details of the variables included in each model for APF and ALF series are presented in
Tables 7 and 8, respectively.

(Table 7)

(Table 8)

It is observed from tables 7 and 8 that the inclusion of multiple variables greatly improves the modelling of hydrological
extremes in comparison to models that use a single variable. It was found in all stations that the final model M8 which
includes PI and LULC variables is the best model to explain the variability of APF and ALF, followed by the M5 model
which includes PI, and by the M4 model which uses CI (with the exception of ALF in Surucucho where M7 which uses CI
and LULC was the second and M5 the third in performance).

According to the M4 model, large-scale climate indices which drive the dynamics and the occurrence of APF in Surucucho
are related with ENSO and explained by TNI and ENSO12 indices; while for Matadero, ENSO through the TNI, and CAR
indices modulate the dynamics of APF. On the other hand, the ALF in Surucucho is driven by PDO and AMO. In Matadero,
ALF is related to ENSO through ENSO34 and ENSO12, and with TSA and PDO dynamics.

Model M8 shows that the variability of APF in Surucucho depends on climate variability only, while the variability of APF
in the Matadero station is driven by climate and LULC changes (in some extent). In Surucucho, the APF variability is related
to AP3 and AP0, the latter variable was not identified in the single models, but it was included in the final model. In
Matadero, M8 shows that the variability of APF is related to AP11 and páramo and forest land use dynamics. Regarding to





ALF, in Surucucho, SPI1_lag1, SPI3_lag2, páramo and forest land uses were retained in the final model; while in Matadero, SPI1_lag1, SPI1_lag0, SPI1_lag2 and páramo land use modulate the variability of ALF series. A comparison between M5 (which use PI variables only) and M8 models show that PI accounts for the major variability of the hydrological extremes;
however, the influence of LULC changes should not be omitted. This is especially true for the Matadero station where greater LULC changes have occurred, and therefore, greater differences between M5 and M8 models are observed in comparison to the Surucucho station according to the AIC values (Table 7 and 8).

Figure 7 shows the diagnostic plots for M8 models. It is observed that M8 models reproduce the dynamics of APF and ALF with the majority of the observed points within the 5-95 % centile curve in all the stations. However, a better representation
of the variability of the hydrological extremes is obtained in Matadero than in Surucucho. Comparing Fig. 5-6 and 7, it is clearly observed that multiple covariate models outperformed single covariate models, explaining in better way the variability of hydrological extremes. In addition, all the worm points of the hydrological extremes in both stations, are within the 95 % confidence intervals, indicating a good fit of the models. The Filliben coefficients (Table 7 and 8) of M8 models were higher than 0.963, showing good performance of the models.
360                                                          (Figure 7)

## 5 Discussion

Results show that annual hydrological extremes present an overall positive trend during the study period; however, different trends were observed for ALP and ALF before and after change points, suggesting a change of hydrological extremes through time. These changes were more evident for APF in Surucucho and Matadero, and for ALF in Surucucho as they
reported significant trends and/or change points. Multiple global climate indices (CI), local climate variables (PI variables) and anthropogenic (LULC) changes were evaluated to investigate which of them mostly control the hydrological extremes behavior, with the final aim to explain the causes of its changing features. The results demonstrate that considering multiple interactions and drivers is necessary for capturing the non-stationary variability of annual hydrological extremes.

Large-scale climate indices such as TNI, ENSO12 and CAR primary explain the variability of APF in our study area. These
results agreed with previous research carried out in the same location (e.g. Campozano et al. 2018; Mendoza et al. 2019; Mora and Willems 2012). For instance, on the one hand, a significant positive correlation between the mesoscale convective systems occurrence and the TNI was found, especially during the rainy season (Campozano et al., 2018; Mendoza et al., 2019). On the other hand, Quishpe-Vásquez et al. (2019) reported that ENSO teleconnection indices, which take into account the temperature gradient between the eastern and western tropical Pacific (i.e. TNI), exhibited significant and stable
correlations with streamflow in the rainy season in the Andean region of Ecuador. Furthermore, Mora and Willems (2012) reported a great positive correlation of precipitation with ENSO12 during March-May in higher regions of the Paute basin; while CAR influence the inter Andean valley (in agreement with the inclusion of CAR in the Matadero station). The last-mentioned is also supported by Mendoza et al. (2019).




On another hand, ENSO34 and ENSO12, TSA, PDO and AMO indices drive ALF in our study area. The inclusion of
ENSO34, ENSO12 and TSA are in accordance with previous studies in the Andes region; although, the influence of PDO
and AMO on ALF are interesting findings. According to Vicente-Serrano et al. (2017), the ENSO34 index is better related to
drought variability in the Andes than ENSO12. More recently, Quishpe-Vásquez et al. (2019) also reported that streamflow
during the dry season in the Andes is strongly correlated to the ENSO34 index, which is in accordance with our findings.
Although several studies have supported the superiority of the ENSO34 index to describe the variability of precipitation in
the Andes over the ENSO12 index (e.g. Morán-Tejeda et al. 2016; Tobar and Wyseure 2018; Vicente-Serrano et al. 2017),
our results showed that both indices (i.e. ENSO34 and ENSO12) are complementary to describe ALF in the studied area (as
was observed for the Matadero station). In fact, Mora and Willems (2012) found greater negative correlations of
precipitation with ENSO12 and positive correlation with TSA during dry seasons.

Focusing on a local climate scale, it was found that climate factors are the main cause of variability in the dynamics of both
hydrological extremes. It is important to note that most of the APF events are mainly related to cumulative volumes of
precipitation of several days prior to the events, rather than to the precipitation of the same day of the event. In Surucucho,
AP of 3 days and 0 day drives APF, while AP of 11 days, 3 days and 0 day controls the annual peaks in Matadero. Analyzing
the AP series included in the final models of APF at both stations, a positive trend is observed in both variables similar to the
APF series. Although no statistically significant trends or changes points were observed for AP, the positive trends and
change points around 1991-1992 identified in AP series similar to the APF, suggest that APF increased as a consequence of
the increase in precipitation. Increasing precipitation and streamflow trends have been previously reported in the Andes for
the wet season (e.g. Celleri et al. 2007; Morán-Tejeda et al. 2016; Quishpe-Vásquez et al. 2019; Tobar and Wyseure 2018).
Besides precipitation trends, the decreasing in páramo and forest land cover seems to exacerbate the magnitude and
variability of APF in Matadero. An extensive literature review showed that Andean forests intercept around 25 % of annual
rainfall and are capable to attenuate peak flows even during more intense storm events (Bonnesoeur et al., 2019).
Bonnesoeur et al. (2019) stated that peak flows with a return period of less than 10 years can be significantly reduced by
forest cover. Recently, Ochoa-Sánchez, Crespo, and Célleri (2018) pointed out the great role of páramo interception on the
catchment hydrology. The authors found that for larger events, relative interception represents 10 % of precipitation while
for small events accounts for higher than 80 % of precipitation. A significant change point in 1992 similar to the APF was
detected in both catchments for all LULC types; this could be explained due to the rapid increase of the agricultural frontier
as a consequence of the opening of the Cuenca-Molleturo-Naranjal road which was ended around 1992-1993. However, the
influence of land cover changes could be neglected for Surucucho due to the lower changes in LULC in comparison to the
Matadero station.

Regarding to ALF, changes in precipitation between 1 to 4 months prior to the event were associated to the occurrence of
these extremes at the Surucucho station, while in Matadero, precipitation variability from the event-month to 2 months prior
to the event were included in the final model. These greater lead times in the Surucucho station could be due to the higher
regulation capacity of the catchment because of the relatively greater amount of páramo in Llaviucu than in the Tomebamba

catchment (Table 2). It was found that all the SPI variables included at both stations, showed a decreasing trend since 1984-1992 (results not shown), which are close to the change points identified in ALF at the Surucucho and Matadero stations.

Besides the overall increasing trend of ALF in the study area, a slightly decreasing trend was observed for ALF after the change points. The similar trends of SPI suggest that changes in ALF are produced by the reduction of precipitation in months that precedes ALF extremes. Besides the precipitation, the inclusion of páramo land cover in the final M8 model significantly improves the capability to describe ALF. The greater improvement in the inclusion of páramo dynamics in Matadero might be explained by the bigger changes in this land use than in the Llaviucu catchment (Surrucucho station).

Several studies have shown the importance of páramo land cover. Ochoa-Tocachi et al. (2016) showed that grazing in páramo catchments result in a significant reduction of the catchment regulation capacity and in water yield.

## 6 Conclusions

The causes of streamflow variation for extremes in relation to global climate variability and local hydrological changes were investigated for tropical-mountain systems located in the south of Ecuador. The main conclusions can be summarized as

follows:

On the one hand, the annual-peak-flow (APF) series show an increasing trend during the study period. Additionally, a threshold point is observed in 1992, from which the increasing rate is strengthened. Interestingly, the trending intensity of APF series is more marked in the streamflow of lower lands (i.e. in Matadero station) than for streamflow of higher lands (i.e. Surucucho station). On the other hand, for annual-low-flow (ALF) series, a significant increasing trend is also revealed.

Nonetheless, threshold points from which their rate of change is weakened are present on 1981 and 1986, for Matadero and Surucucho respectively.

The APF variability responds mainly to cumulative precipitation of 3 and 11 previous days for Surucucho and Matadero respectively. The latter is reasonable since cumulative precipitation synthesizes the antecedent moisture conditions in the catchments. Nonetheless, global climate signals underneath local climate plays an important role in controlling the APF non-

stationary behavior (TNI, ENSO12 and CAR indexes). Furthermore, changes in LULC variables (páramo-grassland and forest) seem to strengthen the variability and magnitude of APF.

Similarly, the ALF variability is strongly explained by the Standardized-Precipitation-indexes (SPI with lags of 0,1,2 and 3 months). Again, these results are reasonable since SPI indexes implicitly encode drought information of local climate. Nonetheless, global climate phenomena exert a clear influence in modulating low extremes (particularly, ENSO34 and

ENSO12, PDO, TSA and AMO signals). LULC effects, especially páramo-grassland and forest changes, play an important role in describing the variability of ALF series; although there seems to be more important for the low lands streamflows (Matadero station), for which the LULC disturbances are more intense than in high lands streamflow (Surucucho station).

In summary, for the tropical mountain systems, besides the importance of local hydrological indexes (PI and SPI), the APF and ALF variability are sensible to the underneath climate factors in higher lands (Surucucho), while for lower lands

(Matadero), the land use/land cover (LULC) changes become important. This knowledge could be stepping milestone



supporting hydrological extremes models under the GAMLSSS framework (e.g., Slater et al. 2017). Further studies should focus on analyzing the relation of different lag times between the annual hydrological extremes and large climate indices in order to construct more robust tools to forecast the risk of floods and droughts events (e.g., Zambrano Mera, Rivadeneira Vera, and Pérez-Martín 2018).


**Code availability:** The GAMLSS models were implemented in the R software. The script used in this study is available upon request.

**Data availability:** Data are available from the authors by request.


**Author contribution:** AA and JC designed the research; JP conducted the analysis of CA-Markov; JC conducted the formal analysis and prepared the manuscript; DM and AA provided comments on the analysis; all the authors contributed to the writing and revisions.

**Acknowledgments:** The authors thank to the Ecuadorian Corporation for the Development of Research and Academia (CEDIA) through the project "Evaluación de los efectos de las actividades socioeconómicas en el cambio del uso del suelo y del cambio climático en las amenazas a inundaciones y sequías en la cuenca del río Tomebamba" for its financial support.

**Financial support.** This research has been supported by the CEDIA (grant no. CEPRA XII-2018-08, Cambio Climático).

**Competing interests:** The authors declare that they have no conflict of interest.

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





**Table 1.** Hydrological stations used in the study area.

| Station name | Code | Catchment | Lat. | Lon. | Elevation (m a.s.l) | Drainage area (km$^2$) | Gaps (%) | Streamflow (m$^3$ s$^{-1}$) | Protected area (%) |
|---|---|---|---|---|---|---|---|---|---|
| Surucucho AJ Llullucchas | H0897 | Llaviucu | -2.83859 | -79.12334 | 3046 | 51.54 | 6.1 | 1.14 | 93.8 |
| Matadero en Sayausí | H0896 | Tomebamba | -2.87549 | -79.07297 | 2701 | 299.42 | 3.89 | 6.34 | 43.6 |








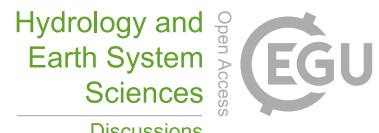


**Table 2.** Percentage of land use and land cover (LULC) in the study catchments.

| Land cover type | Llaviucu | | | Matadero | | |
|---|---|---|---|---|---|---|
| | 1977 | 2008 | Δ 1977-2008 | 1977 | 2008 | Δ 1977-2008 |
| Forest | 7.13 | 7.00 | -0.13 | 9.69 | 7.53 | -2.16 |
| Lagoons | 3.52 | 3.58 | 0.06 | 1.49 | 1.55 | 0.06 |
| Bare land | 0.04 | 0.06 | 0.02 | 0.29 | 0.55 | 0.26 |
| Páramo grassland | 85.86 | 85.00 | -0.86 | 79.74 | 78.45 | -1.29 |
| Grazing/Cultivated land | 2.10 | 2.86 | 0.76 | 5.23 | 8.63 | 3.40 |
| Shrubland | 1.35 | 1.49 | 0.14 | 3.56 | 3.30 | -0.26 |








**Table 3.** Description of the non-stationary models.

| Model code | Model name | Covariates | Construction of models |
|---|---|---|---|
| M0 | Stationary model | - | - |
| M1 | Single CI model | CI | Best single CI covariate |
| M2 | Single PI model | PI | Best single PI covariate |
| M3 | Single LULC model | LULC | Best single LULC covariate |
| M4 | Multiple CI model | CI | Identified by stepwise with all CI as candidates |
| M5 | Multiple PI model | PI | Identified by stepwise with all PI as candidates |
| M6 | Multiple LULC model | LULC | Identified by stepwise with all LULC as candidates |
| M7 | Multiple CI+LULC model | CI+LULC | Identified by stepwise with CI variables of M4 and all LULC variables as candidates |
| M8 | Multiple PI+LULC model | PI+LULC | Identified by stepwise with PI variables of M5 and all LULC variables as candidates |









**Table 4.** Trends (Zc) and change points detection of hydrological extreme series and explanatory variables for the Surucucho and Matadero stations.

| | Annual Peak Flow | | | | | Annual Low Flow | | | |
| --- | --- | --- | --- | --- | --- | --- | --- | --- | --- |
| | Surucucho | | Matadero | | | Surucucho | | Matadero | |
| Variable | Zc | Change point | Zc | Change point | Variable | Zc | Change point | Zc | Change point |
| APF | 1.59 | *1992* | *2.07* | *1992* | ALF | *3.00* | *1986* | 1.57 | 1981 |
| ENSO12 | -0.93 | 2001 | -0.06 | 1990 | ENSO12 | -1.07 | 1994 | -0.93 | 1988 |
| ENSO3 | 0.21 | 1985 | -0.25 | 2000 | ENSO3 | -0.55 | 1997 | -0.09 | 2001 |
| ENSO34 | 0.73 | 1985 | -0.18 | 1997 | ENSO34 | -0.02 | 1997 | 1.12 | 2000 |
| ENSO4 | 1.21 | 1988 | 0.43 | 1989 | ENSO4 | 1.32 | 1989 | *2.02* | *1989* |
| TNI | -1.07 | 2000 | -0.89 | 2000 | TNI | *-2.64* | *1989* | *-2.85* | *2000* |
| MEI | -0.86 | 1994 | -0.89 | 1997 | MEI | -0.46 | 1997 | -0.07 | 1997 |
| ONI | -0.73 | 1997 | -0.89 | 1997 | ONI | -0.39 | 1997 | 0.04 | 1989 |
| SOI | 1.14 | 1994 | 0.84 | 1997 | SOI | -0.07 | 1989 | -0.89 | 1989 |
| PDO | -1.25 | 1987 | -0.89 | 1997 | PDO | -1.46 | 1997 | *-2.63* | *1997* |
| AMO | *2.62* | *1995* | *4.32* | *1994* | AMO | *3.21* | *1994* | *3.57* | *1994* |
| AMM | 0.79 | 1999 | 1.25 | 1994 | AMM | *2.25* | *1994* | 1.50 | 1994 |
| NAO | -0.18 | 1996 | *-2.35* | 1994 | NAO | 0.37 | 1984 | -0.73 | 1995 |
| TNA | 1.50 | **1994** | *1.89* | *1994* | TNA | *2.69* | *1994* | *2.57* | *1994* |
| TSA | 1.55 | 1982 | 1.55 | 1982 | TSA | 1.18 | 1983 | 1.64 | 1983 |
| CAR | *2.54* | *1996* | *3.63* | *1996* | CAR | *3.23* | *1993* | *3.55* | *1993* |
| QBO | -0.57 | 1990 | 0.32 | 1995 | QBO | -0.82 | 1999 | -0.79 | 1995 |
| AP0 | 0.11 | 1988 | -0.21 | 1997 | SPI1_lag0 | 0.42 | 1986 | 0.14 | 1984 |
| AP1 | -0.07 | 1992 | -0.64 | 1995 | SPI1_lag1 | 0.93 | 1984 | 0.39 | 1982 |
| AP2 | 0.21 | 1992 | -0.57 | 1995 | SPI1_lag2 | 0.32 | 2000 | 0.45 | 1986 |
| AP3 | 0.68 | 1992 | -0.33 | 1982 | SPI1_lag3 | 0.61 | 1986 | -0.04 | 1996 |
| AP4 | 0.36 | 1992 | -0.57 | 1982 | SPI1_lag4 | 0.68 | 1992 | 0.79 | 1994 |
| AP5 | 0.32 | 1992 | -0.29 | 1983 | SPI1_lag5 | -0.75 | 2002 | -0.43 | 2000 |
| AP6 | 0.21 | 1992 | -0.18 | 1983 | SPI1_lag6 | **1.86** | 1995 | 0.14 | 1992 |
| AP7 | -0.04 | 1983 | -0.14 | 1983 | SPI3_lag0 | 0.39 | 1981 | 0.25 | 2000 |
| AP8 | 0.07 | 2005 | -0.11 | 1983 | SPI3_lag1 | 1.28 | 1994 | 1.50 | 1992 |
| AP9 | 0.36 | 1995 | 0.00 | 1983 | SPI3_lag2 | 1.13 | 1981 | 1.39 | 1993 |
| AP10 | 0.00 | 1985 | 0.39 | 1983 | SPI3_lag3 | 0.75 | 1992 | 0.11 | 1993 |
| AP11 | -0.39 | 1983 | 0.68 | 1991 | SPI3_lag4 | **1.71** | **1992** | 0.18 | 1993 |
| AP12 | -0.18 | 1983 | 0.96 | 1991 | Forest | *-3.90* | *1992* | *-3.84* | *1992* |
| AP13 | -0.25 | 1983 | 0.93 | 1991 | Páramo | *-3.68* | *1992* | *-4.39* | *1992* |
| AP14 | -0.18 | 1983 | 0.89 | 1991 | Grazing | *3.60* | *1992* | *4.37* | *1992* |
| Forest | *-3.90* | *1992* | *-3.84* | *1992* | | | | | |
| Páramo | *-3.68* | *1992* | *-4.39* | *1992* | | | | | |
| Grazing | *3.60* | *1992* | *4.37* | *1992* | | | | | |

Note: In bold significant trends/change points at 10% significance level, in bold and italics significant trends/change points at 5% significance level.





**Table 5.** Summary of significant covariates included in the single covariate models for modelling APF in Surucucho and Matadero. Models are ranked according to the AIC-performance.

| | Surucucho | | | | | Matadero | | | |
|---|---|---|---|---|---|---|---|---|---|
| Ranking | Model | $\theta_1$ | $\theta_2$ | $\theta_3$ | AIC | Model | $\theta_1$ | $\theta_2$ | AIC |
| 1 | M2_AP3 | AP3 | cte | cte | 120.20 | M2_AP11 | AP11 | cte | 240.79 |
| 2 | M2_AP4 | AP4 | cte | cte | 121.26 | M2_AP10 | AP10 | cte | 241.22 |
| 3 | M2_AP5 | AP5 | cte | cte | 121.39 | M2_AP12 | AP12 | cte | 241.32 |
| 4 | M1_TNI | TNI | TNI | cte | 121.73 | M1_TNI | cte | TNI | 242.27 |
| 5 | M2_AP6 | AP6 | cte | cte | 122.74 | M2_AP13 | AP13 | cte | 242.30 |
| 6 | M2_AP2 | AP2 | cte | cte | 123.10 | M2_AP9 | AP9 | cte | 244.44 |
| 7 | M2_AP7 | AP7 | cte | cte | 124.07 | M2_AP14 | AP14 | cte | 244.51 |
| 8 | M1_MEI | MEI | MEI | cte | 124.87 | M2_AP8 | AP8 | cte | 245.56 |
| 9 | M1_ENSO12 | cte | ENSO12 | cte | 125.03 | M1_ENSO12 | cte | ENSO12 | 245.90 |
| 10 | M2_AP8 | AP8 | cte | cte | 125.31 | M2_AP7 | AP7 | cte | 246.27 |
| 11 | M2_AP1 | AP1 | cte | cte | 125.74 | M1_CAR | CAR | CAR | 246.30 |
| 12 | M2_AP9 | AP9 | cte | cte | 125.74 | M2_AP6 | AP6 | cte | 246.44 |
| 13 | M2_AP10 | AP10 | cte | cte | 126.27 | M2_AP5 | AP5 | cte | 246.94 |
| 14 | M2_AP11 | AP11 | cte | cte | 126.48 | M3_Grazing | Grazing | cte | 247.36 |
| 15 | | | | | | M3_Forest | Forest | cte | 247.38 |
| 16 | | | | | | M3_Páramo | Páramo | cte | 247.58 |
| 17 | | | | | | M2_AP1 | AP1 | cte | 250.59 |
| 18 | | | | | | M2_AP2 | AP2 | cte | 247.82 |
| 19 | | | | | | M1_ENSO4 | cte | ENSO4 | 247.90 |



**Table 6.** Summary of significant covariates included in the single covariate models for modelling ALF in Surucucho and Matadero. Models are ranked according to the AIC-performance.

| | Surucucho | | | | Matadero | | | |
|---|---|---|---|---|---|---|---|---|
| Ranking | Model | $\theta_1$ | $\theta_2$ | AIC | Model | $\theta_1$ | $\theta_2$ | AIC |
| 1 | M3_Páramo | Páramo | cte | -34.29 | M2_SPI1_lag1 | SPI1_lag1 | cte | 78.33 |
| 2 | M3_Grazing | Grazing | cte | -34.25 | M2_SPI1_lag0 | SPI1_lag0 | cte | 85.16 |
| 3 | M3_Forest | Forest | cte | -32.76 | M1_ENSO3 | ENSO3 | ENSO3 | 88.05 |
| 4 | M2_SPI3_lag2 | cte | SPI3_lag2 | -32.52 | | | | |
| 5 | M1_AMM | cte | AMM | -31.66 | | | | |
| 6 | M2_SPI1_lag1 | SPI1_lag1 | cte | -31.53 | | | | |
| 7 | M1_PDO | PDO | PDO | -31.18 | | | | |
| 8 | M2_SPI3_lag0 | SPI3_lag0 | cte | -30.53 | | | | |
| 9 | M1_ENSO4 | ENSO4 | cte | -30.31 | | | | |
| 10 | M2_SPI3_lag1 | cte | SPI3_lag3 | -30.25 | | | | |
| 11 | M1_TNA | TNA | cte | -30.22 | | | | |
| 12 | M2_SPI1_lag2 | cte | SPI1_lag2 | -30.20 | | | | |
| 13 | M2_SPI1_lag5 | SPI1_lag5 | SPI1_lag5 | -30.02 | | | | |
| 14 | M1_AMO | AMO | cte | -29.93 | | | | |
| 15 | M2_SPI1_lag0 | SPI1_lag0 | cte | -29.72 | | | | |
| 16 | M1_ENSO12 | cte | ENSO12 | -29.41 | | | | |
| 17 | M1_CAR | CAR | cte | -28.92 | | | | |






**Table 7.** Summary of significant covariates included in the best single covariate and multiple covariate models and their Filliben coefficient values to model APF in Surucucho and in Matadero. Models are ranked according the AIC-performance.

| | Surucucho | | | | | Matadero | | | | |
|---|---|---|---|---|---|---|---|---|---|---|
| Ranking | Model | $\theta_1$ | $\theta_2$ | $\theta_3$ | AIC | Filliben | Model | $\theta_1$ | $\theta_2$ | AIC | Filliben |
| 1 | M8 (M5) | AP3 AP0 | cte | cte | 116.44 | 0.991 | M8 | AP11 Forest Páramo | Forest | 226.25 | 0.993 |
| 2 | M7 (M4) | TNI | ENSO12 | cte | 118.55 | 0.992 | M5 | AP11 | AP3 AP0 | 230.18 | 0.996 |
| 3 | M2 | AP3 | cte | cte | 120.20 | 0.989 | M7 (M4) | CAR | TNI | 238.77 | 0.980 |
| 4 | M1 | TNI | TNI | cte | 121.73 | 0.992 | M2 | AP11 | cte | 240.79 | 0.992 |
| 5 | M0 | cte | cte | cte | 128.80 | 0.987 | M1 | cte | TNI | 242.27 | 0.974 |
| 6 | | | | | | | M6 (M3) | Grazing | cte | 247.36 | 0.987 |
| 7 | | | | | | | M0 | cte | cte | 249.97 | 0.986 |









**Table 8.** Summary of significant covariates included in the best single covariate and multiple covariate models and their Filliben coefficient values to model ALF in Surucucho and in Matadero. Models are ranked according to the AIC-performance.

| | Surucucho | | | | | Matadero | | | | |
|---|---|---|---|---|---|---|---|---|---|---|
| Ranking | Model | $\theta_1$ | $\theta_2$ | AIC | Filliben | Model | $\theta_1$ | $\theta_2$ | AIC | Filliben |
| 1 | M8 | SPI1_lag1 Páramo Forest | SPI3_lag2 | -43.17 | 0.986 | M8 | SPI1_lag1 SPI1_lag0 SPI1_lag2 | Páramo | 67.48 | 0.981 |
| 2 | M5 | SPI1_lag1 | SPI3_lag2 | -40.71 | 0.993 | M7 | ENSO34 TSA | ENSO12 Páramo Forest | 67.55 | 0.988 |
| 3 | M7 | Páramo | PDO | -39.06 | 0.994 | M5 | SPI1_lag1 SPI1_lag0 SPI1_lag2 | cte | 74.89 | 0.986 |
| 4 | M4 | AMO | PDO | -34.51 | 0.994 | M4 | ENSO34 TSA | ENSO12 TSA PDO | 75.86 | 0.976 |
| 5 | M3 (M6) | Páramo | cte | -34.29 | 0.987 | M2 | SPI1_lag1 | cte | 78.33 | 0.992 |
| 6 | M2 | cte | SPI3_lag2 | -32.52 | 0.985 | M6 | Páramo Grazing Forest | cte | 83.44 | 0.986 |
| 7 | M1 | cte | AMM | -31.66 | 0.981 | M4 | ENSO3 | ENSO3 | 88.05 | 0.984 |
| 8 | M0 | cte | cte | -26.95 | 0.989 | M0 | cte | cte | 90.97 | 0.990 |






**Figure 1.** Location of the study area and hydrological and rain gauge stations used in the study. The 30 m resolution digital elevation model (DEM) in the map was created by the National Aeronautics and Space Administration (NASA) and its Shuttle Radar Topography Mission (SRTM) project, which is available in the web page of the Instituto Geográfico Militar of Ecuador (IGM; http://www.geoportaligm.gob.ec/portal/).



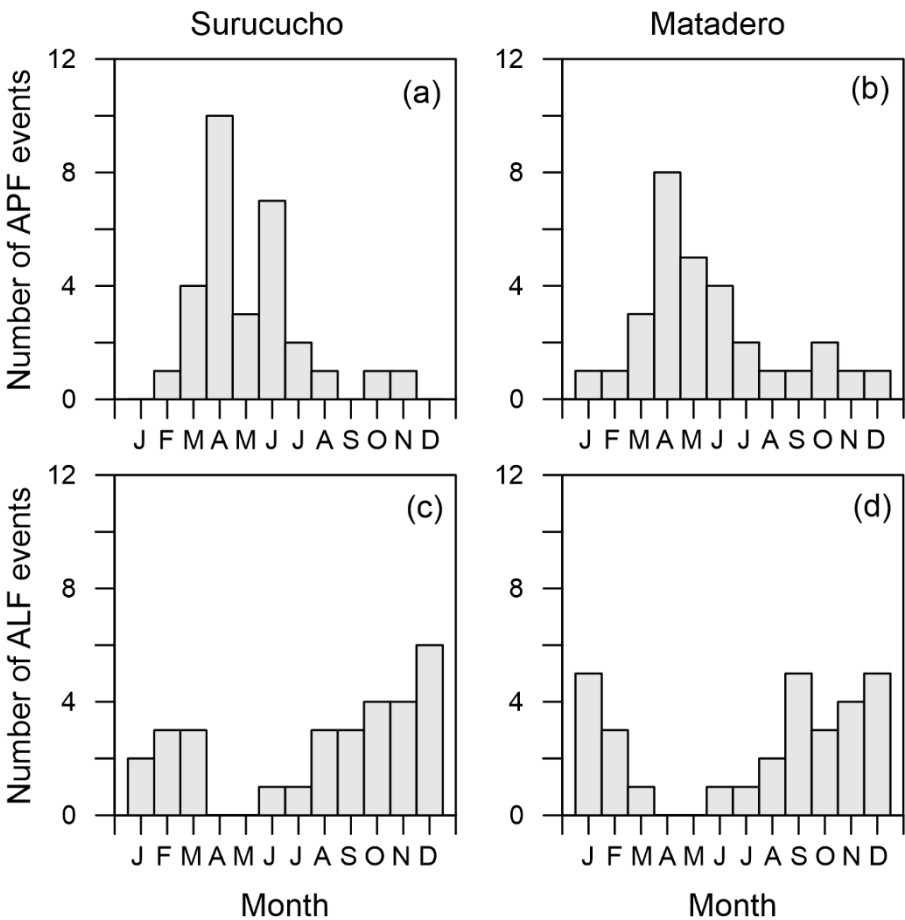

**Figure 2.** Monthly distribution of annual hydrological extremes during 1978-2007. Annual peak flows (APF; a-b) and annual low flows (ALF; c-d). The Surucucho station on the left panel and the Matadero station on the right panel.

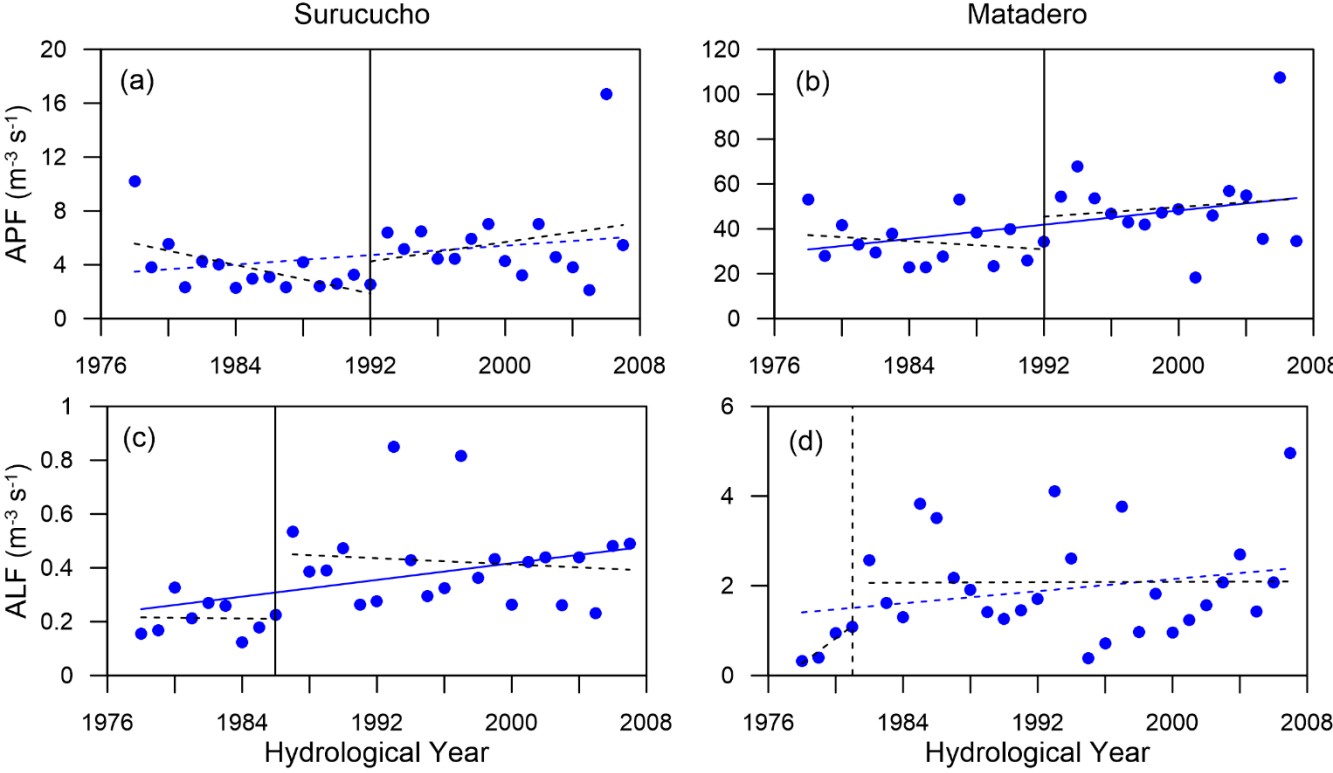

**Figure 3.** Annual peak flow series (APF; a-b) and annual low flow series (ALF; c-d) during 1978-2007, the Surucucho
station on the left hand panel and the Matadero station on the right hand panel. Continuous and dash lines indicate significant
(p-value < 0.05) and no significant (p-value ≥ 0.05) trends/change points, respectively.



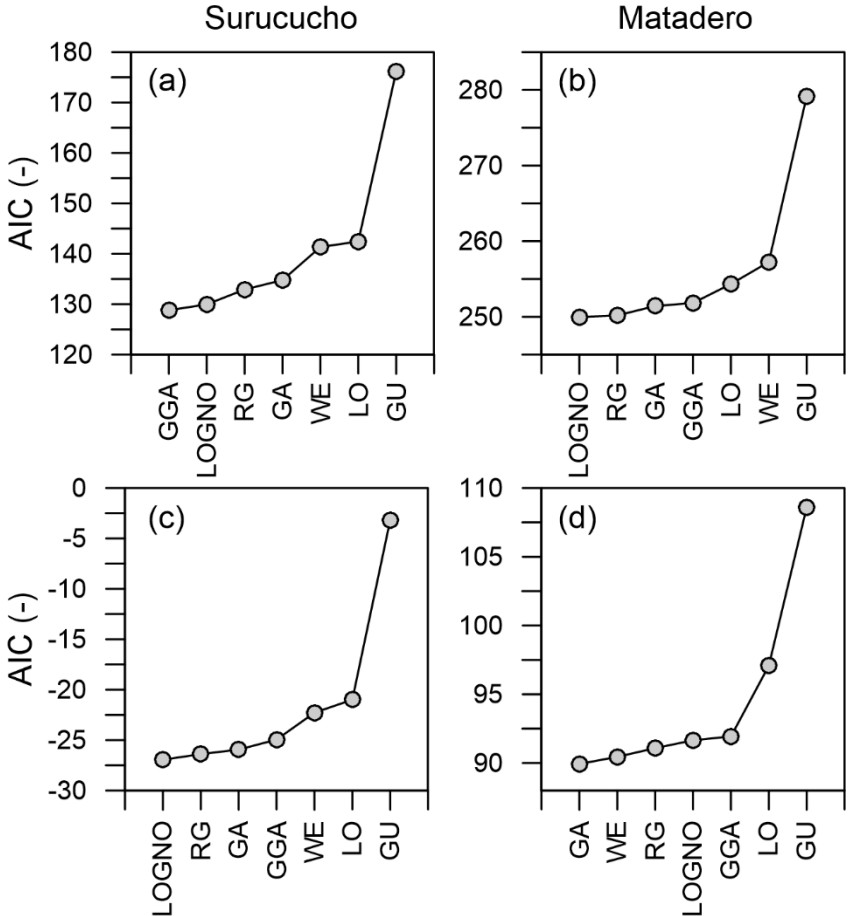

**Figure 4.** Fitting results of the APF (a-b) and ALF (c-d) series under stationary conditions. AIC values in Surucucho station on the left panel and Matadero station on the right panel. Note that the models are ranked in order of performance.







**Figure 5.** Evaluation of the APF series for the best single models in Surucucho (a) and in Matadero (b). Upper panels are the centile curves plots and the lower panels are worm plots.



**Figure 6.** Evaluation of the ALF series for the best single models in Surucucho (a) and in Matadero (b). The upper panels are the centile curves plots and the lower panels are worm plots.

890

895









900

**Figure 7.** Diagnostic (upper panels) and worm (lower panels) plots for the M8 model for APF (a-b) and ALF (c-d) series. Surucucho on the left panel and Matadero on the right panel.