# Peer review of "The influence of global climate and local hydrological variations over streamflow extremes: The tropical-mountain case"

_Hydrology and Earth System Sciences, 2019_

## Referee Comment (RC1) · Anonymous Referee #1 · 21 Jan 2020

This manuscript assesses the contribution of several factors, including precipitation factor, land use change and large scale climate indices on hytdrological extreme change, using the statistical approach. My major worry is that work about statistically investigating the influences from different drivers on hydrological extremes is not new, and the data/tools used by the authors are also conventional. In this condition the authors should explicitly illustrate their differences in findings and interpretation by comparing to different former studies. Nevertheless, this part is still weak. The authors might try to make their findings are representable to different areas as they stated that their study area is natural laboratory for hydrological and climate research. However, I found their statements in introduction too focus on the Andean area. This might restirct the global

significance of their work and tend to make it like a regional study.

---

## Referee Comment (RC2) · Anonymous Referee #2 · 15 Feb 2020

First, I have to say that I am attracted by the title of the manuscript before accepting to review this manuscript. However, after I carefully went through the manuscript, it is definitely not what I thought that focus on the physical influence of global and local drivers to streamflow extremes, so I think the title should be more specific on the basins or some more related to regional studies.

This manuscript uses the GAMLSS model to analyzed the nonstationarity of streamflow extremes over two stations. Frankly speaking, both the method and nonstationarity related to the large-scale climate variability are very common for many previous studies. I personally used GAMLSS model to study the nonstationarity of Canadian floods with

more than 100 stations (Tan et al., 2015). This paper focuses on only two stations and examined only statistical relations between streamflow extremes and climate indices. As I understanding, the relations detected might be only statistical but without any physical reasons, therefore, I think the authors should be more looking at some physical mechanisms. Therefore, I suggest the authors make substantial improvements on the way to be publication. The following are some commentsïijŇ

(1) The authors used too many climate indices. Since many climate indices used have strong correlations, so I think it is not necessary to use a variety of climate indices, without previous selection based on the physical relations between global climate and region hydrology. Again, some statistical relations can only be statistical, but no real meaning to promote understanding of teleconnections and predictability of regional hydrology. (2) The treatment of change points in statistical analyses. The authors detected change points for both time series of streamflow extremes over two basins. Whether the change points are due to the nonlinear relation between climate indices and streamflow extremes? The nonlinear relations are very common in teleconnections, even though I do not know this exists in South America and large scale climate variability, but it quite is evident in Australia, e.g. Cai et al., 2012 and 2013. So how do the authors consider the change points in GAMLSS analyses? (3)This study only examined two time series, which make me thought that the study should more focus on the physical teleconnection but not statistical relations, because of the limited samples. (4)To make the study more attractive, the manuscript should point out the novelty of GAMLSS analyses. Currently I do not see new points. (5)The authors implemented precipitation information to GAMLSS model. Here, I think the authors should specify the purpose of using precipitation information to predict streamflow extremes. The relations between precipitation and streamflow is quite straightforward and there is no need to use GAMLSS model to find this relation. Moreover, because the precipitation and streamflow are generally not lag-correlated but changes simultaneously on time scales larger than monthly, so preciptation do not provide any predictability to streamflow extremes, even though the relations can be found by some statistical analyses.
Minor comments Line 80: both hydrological extremes? I think should be extremely low and high streamflow? There is should be Is there? Line 265: they were not significant.

Cai, W., and P. van Rensch, 2013: Austral Summer Teleconnections of Indo-Pacific Variability: Their Nonlinearity and Impacts on Australian Climate. Journal of Climate, 26, 2796-2810. Cai, W., P. van Rensch, T. Cowan, and H. H. Hendon, 2012: An Asymmetry in the IOD and ENSO Teleconnection Pathway and Its Impact on Australian Climate. Journal of Climate, 25, 6318-6329. Tan, X., and T. Y. Gan, 2015: Nonstationary analysis of annual maximum streamflow of Canada. Journal of Climate, 28, 1788-1805.

---

## Referee Comment (RC3) · Anonymous Referee #3 · 23 Feb 2020

First of all, please allow me to make a suggestion. The current form of the submission is not friendly to readers, which put all the tables and figures at the end of the paper. Maybe this is required by the journal in the stage of submission, but I hope next time we can get a better version with all the tables and figures in the text. This will make the reading much easier.

Hydrological extremes such as floods and droughts are occurs more frequently with the climate change and human activities. The study of "The influence of global climate and local hydrological variations over streamflow extremes: The tropical-mountain case " is of great interest. The paper is well written. My suggestion is: Minor Correction.

There are some minor comments: 1.Further strengthen the novelties of this work in the Abstract and text. Clarify the differences and improvements of this work compared with the previous work. 2.Lines 81-82, "(3) There is a difference in the mechanisms of control of hydrological extremes in a relatively undisturbed and disturbed catchment? ", is this sentence supposed to be a question? 3.Lines 212, "Is important to note that linear approximations are the first terms within Tylor's expansions and generally...", the subject is missing. I think it should be change to "It is important to note that linear approximations are the first terms within Tylor's expansions and generally..."". 4.Line 231, "explanatory variables influence the performance of the models to describe non-stationary ", should be changed to "explanatory variables influence the performance of the models to describe non-stationarity" 5.Figure 3, the legend is missing. The caption is not clear enough for readers to understand the figure.

---

## Author Comment (AC1) · 11 Apr 2020

1.- This manuscript assesses the contribution of several factors, including precipitation factor, land use change and large scale climate indices on hydrological extreme change, using the statistical approach. My major worry is that work about statistically investigating the influences from different drivers on hydrological extremes is not new, and the data/tools used by the authors are also conventional. In this condition the authors should explicitly illustrate their differences in findings and interpretation by comparing to different former studies. Nevertheless, this part is still weak.

Thanks for the comments. It is well-known that the Andean-region plays a fundamental

natural role, and yet, full monitoring systems are still scarce, which limits the applicability of more sophisticated techniques for more in-depth research. However, the more conventional GAMLSS technique can tackle essential research questions, making use of conventional – but key – data for obtaining reliable results. These more straightforward but still powerful methods, capable of unveiling relations between local hydrological variables and global climate conditions, constitutes a milestone for future research in the Andean region. The authors highlight this statement and the novelty of our study in Introduction in Lines: 46-56, 69-76, 90-92, 97-104, and 107-108. Also, we clarify the differences in findings and interpretation of results, comparing our study with former research for enhancing its importance. The latter mentioned was included in the Discussion section (Lines: 553-564).

2.- The authors might try to make their findings are representable to different areas as they stated that their study area is natural laboratory for hydrological and climate research. However, I found their statements in introduction too focus on the Andean area. This might restrict the global significance of their work and tend to make it like a regional study.

The reviewer is right in the sense that the results represent expressly to the Andean region. When we pointed out to the location as a "natural laboratory", we refer to the combination of specific very complex climate and hydrological conditions, which makes of it crucial for research under such circumstances. However, we understand that the way we have stated this idea could lead to confusion. Therefore, let us clarify the "natural laboratory" statement. On the one hand, small monitored areas with 30 years of temporal data-sets constitute in a luxury hydrological information for the Andean-systems. Particularly, the monitored nested hydrological catchments – one undisturbed and the other altered – provide the opportunity for contrasting the hydrological reactions that similar climate effects exert over such different systems. On the other hand, since the local information encompass long-term dynamical land cover evolution, it was possible to discern the impact that these cover trends have over low and high extremes, and differencing them from the climate effects. In virtue of the latter mentioned, the authors' statement of "natural laboratory" is used, and we believe that this is where the importance of the study mainly relies on. We clarify all the mentioned in the Introduction in Lines 108-109 and 114-118. Also, we changed the title of the manuscript to the aim of restricting our study to Andean regions.

The revised manuscript is in the following link https://drive.google.com/drive/folders/1YrUG7fMXBhkqCHs-8dpemt72aMf_HCj_?usp=sharing

––––––––––––––––––––––––––––––

---

## Author Comment (AC2) · 11 Apr 2020

I.- First of all, please allow me to make a suggestion. The current form of the submission is not friendly to readers, which put all the tables and figures at the end of the paper. Maybe this is required by the journal in the stage of submission, but I hope next time we can get a better version with all the tables and figures in the text. This will make the reading much easier.

We take the suggestion of the reviewer, in the new version we included the tables and figures in the text. The result is a friendlier version of the manuscript.
II.- Hydrological extremes such as floods and droughts are occurs more frequently with the climate change and human activities. The study of "The influence of global climate and local hydrological variations over streamflow extremes: The tropical-mountain case" is of great interest. The paper is well written. My suggestion is: Minor Correction.

Thanks for the comments, we are glad to hear that the reviewer found our manuscript interesting and well written.

III.- There are some minor comments: 1. Further strengthen the novelties of this work in the Abstract and text. Clarify the differences and improvements of this work compared with the previous work.

We clarify the novelty of our work in Introduction in Lines: 46-56, 69-76, 90-92, 97-104, and 107-108; and in the Abstract in Lines: 14-17, 23 and 28-29.

2. Lines 81-82, "(3) There is a difference in the mechanisms of control of hydrological extremes in a relatively undisturbed and disturbed catchment?", is this sentence supposed to be a question?

It is correct, we declare this sentence as a question. We improved the writing in the text to avoid misinterpretations (Lines 113-114).

3. Lines 212, "Is important to note that linear approximations are the first terms within Tylor's expansions and generally...", the subject is missing. I think it should be change to "It is important to note that linear approximations are the first terms within Tylor's expansions and generally...".

Done. Line 268.

4.Line 231, "explanatory variables influence the performance of the models to describe nonstationary", should be changed to "explanatory variables influence the performance of the models to describe non-stationarity".

Done. Line 288.

5. Figure 3, the legend is missing. The caption is not clear enough for readers to understand the figure.

We added a legend in the Figure 3 to improve the understanding of the readers.

The revised manuscript is in the following link https://drive.google.com/drive/folders/1YzNicBRA4Yi03fekHoVjRfLg4LPeAJh7?usp=sharing

---

## Author Comment (AC3) · 11 Apr 2020

I.- First, I have to say that I am attracted by the title of the manuscript before accepting to review this manuscript. However, after I carefully went through the manuscript, it is definitely not what I thought that focus on the physical influence of global and local drivers to streamflow extremes, so I think the title should be more specific on the basins or some more related to regional studies.

Thanks for the comments. We agree with the suggestion of the reviewer. Therefore, we have changed the last title for a more specific alternative which reflects better the scope and the central research idea. The new title is "The influence of global climate

and local hydrological variations over streamflow extremes: A nested basin case in the Ecuadorian Andes".

II.- This manuscript uses the GAMLSS model to analyzed the nonstationarity of streamflow extremes over two stations. Frankly speaking, both the method and nonstationarity related to the large-scale climate variability are very common for many previous studies. I personally used GAMLSS model to study the nonstationarity of Canadian floods with more than 100 stations (Tan et al., 2015). This paper focuses on only two stations and examined only statistical relations between streamflow extremes and climate indices. As I understanding, the relations detected might be only statistical but without any physical reasons, therefore, I think the authors should be more looking at some physical mechanisms. Therefore, I suggest the authors make substantial improvements on the way to be publication. The following are some comments:

(1) The authors used too many climate indices. Since many climate indices used have strong correlations, so I think it is not necessary to use a variety of climate indices, without previous selection based on the physical relations between global climate and region hydrology. Again, some statistical relations can only be statistical, but no real meaning to promote understanding of teleconnections and predictability of regional hydrology.

It is true, many climate indices have strong correlations. However, we have some comments supporting the methodology and our experimental considerations. Respecting to the correlations between global climate indices, we mentioned that, the GAMLSS technique inherently includes a selection methodology, identifying variables with complementary information into the model, as supported by Figure 1 below. In that sense, when the climate indices are highly correlated, the statistically dominant one will prevail. Although the relations found are only statistically significant – without meaning necessarily causality – the omission of some of them must be supported by scientific evidence contradicting the statistical significance of it. Unfortunately, the dominant climate mechanisms have not been clarified yet for the case of the tropical Andeanregions. In fact, to the authors' knowledge, the evidence of climate connection for these areas show spatially scatter relations, thus suggesting a complex combination of several global climatic effects, rather than a few dominant climatic effects (e.g. Campozano et al., 2018; Mendoza et al., 2019; Mora & Willems, 2012; Vuille et al., 2000). Therefore, we keep with the selected global variables as hypothetical climate drivers exerting an influence on the local hydrological conditions. Nonetheless, the task of unveiling the different underlying climate mechanisms can only be tentatively addressed to support the findings herein. Still, a more comprehensive analysis is out of the scope of our study. We mention all of these in the manuscript over the Methodology section (Lines 166-172). Also, we discuss the results in an attempt to support it and the conclusions based on available literature of the region, this is detailed in Discussion in Lines 464-468, 471-478, 482-493, 496-502 and 505-509.

Figure 1: Correlation matrix of climate indices for hydrological extremes at both stations. The (X) represents the correlations between the selected climate indices, attained through GAMLSS model M4.

(2) The treatment of change points in statistical analyses. The authors detected change points for both time series of streamflow extremes over two basins. Whether the change points are due to the nonlinear relation between climate indices and streamflow extremes? The nonlinear relations are very common in teleconnections, even though I do not know this exists in South America and large scale climate variability, but it quite is evident in Australia, e.g. Cai et al., 2012 and 2013. So how do the authors consider the change points in GAMLSS analyses?

Indeed, nonlinear relations between climate-indices probably play significant roles concerning changing points in extreme flow time-series. There are specialized techniques able to consider significant changes into the GAMLSS framework (i.e. Tan & Gan (2015)). Nonetheless, linear approximations considering here for modelling the GAMLSS's parameters seems to be a rational approach since the linearity is dominant for most of the plots between global indices and streamflow extremes (as shown in Fig.

2 and 3 below). In that sense, multiple-linear regressions could represent the mutual complementary interaction between multiple climate signals handling the extremes-distributions. Of course, severe asymmetries on global climate and local extremes relations entails biased extreme-parameters' estimates by such linear approximations, with a consequence of biased estimations on the entire extreme distributions. The same comment could be applied concerning the role that land cover changes have over these extremes. Hydrological processes have a strong nonlinear nature, which could be enhanced joint with climate nonlinearities. In that sense, linear approximations have the aim of determining simpler, but essential, direct relations as a first approximation analysis of the non-stationary influences that climate and local conditions has over the extremes, but recognizing the limitations of the method and our considerations. Thus, the above mentioned are included as important comments about the considerations into the Methodology in Lines 266-268 and Discussion in Lines 510-518.

Figure 2. Scatter plot of the relationship between annual peak flow (APF) series and climate indices selected in the M4 models at (a) Surucuho and (b) Matadero.

Figure 3. Scatter plot of the relationship between annual low flow (ALF) series and climate indices selected in the M4 models at (a) Surucuho and (b) Matadero.

(3) This study only examined two time series, which make me thought that the study should more focus on the physical teleconnection but not statistical relations, because of the limited samples.

The authors agree about that, due to small samples used herein, it is necessary to support the mechanisms in somehow. Fortunately, there is some specific knowledge about climate mechanisms explaining the relations between climate variables and local rainfall variabilities in the studied area (e.g. Campozano et al., 2018). This evidence, to some extent, supports the causality of our statistical findings. Therefore, based on the latter specific study, we provide some statements explaining how such underlying climate mechanisms involving some of the climate variables and rainfall, have consequences on hydrological extreme's effects. This climate attribution analysis is included in the Discussion section (Lines 471-478).

(4) To make the study more attractive, the manuscript should point out the novelty of GAMLSS analyses. Currently I do not see new points.

We clarify the novelty of our work in Introduction in Lines: 46-56, 69-76, 90-92, 97-104, and 107-108.

(5) The authors implemented precipitation information to GAMLSS model. Here, I think the authors should specify the purpose of using precipitation information to predict streamflow extremes. The relations between precipitation and streamflow is quite straightforward and there is no need to use GAMLSS model to find this relation. Moreover, because the precipitation and streamflow are generally not lag-correlated but changes simultaneously on time scales larger than monthly, so precipitation do not provide any predictability to streamflow extremes, even though the relations can be found by some statistical analyses.

Although precipitation and streamflow relations are well known and evident for several regions (especially for arid or semi-arid areas), these have not been well understood in tropical zones and wetlands areas, mainly because of the different non-stationarity hydrological processes (Buytaert & Beven, 2011). Thus, non-stationary methods (such as GAMLSS) could be helpful to address these non-stationary proprieties. However, the results obtained based on our experimental consideration (i.e. few days of antecedent cumulative precipitation for annual peak flows, and one-month lag precipitation for annual low flows as the main drivers), support the fact that precipitation makes no contribution to any prediction attempt under the GAMLSS framework. We clarify our statements in Introduction in Lines 107-108, in Methodology in Lines 183-187, and in Discussion in Lines 564-568.

(6) Minor comments Line 80: both hydrological extremes? I think should be extremely low and high streamflow? There is should be Is there? Line 265: they were not signifi-

cant.

It is correct, we refer to low streamflow extremes and high streamflow extremes. Both comments were corrected in Lines 112 and 325.

References

Buytaert, W., & Beven, K. (2011). Models as multiple working hypotheses: Hydrological simulation of tropical alpine wetlands. Hydrological Processes, 25(11), 1784–1799. https://doi.org/10.1002/hyp.7936 Campozano, L., Trachte, K., Célleri, R., Samaniego, E., Bendix, J., Albuja, C., & Mejia, J. F. (2018). Climatology and Teleconnections of Mesoscale Convective Systems in an Andean Basin in Southern Ecuador: The Case of the Paute Basin. Advances in Meteorology, 2018. https://doi.org/10.1155/2018/4259191 Mendoza, D. E., Samaniego, E. P., Mora, D. E., Espinoza, M. J., & Campozano, L. V. (2019). Finding teleconnections from decomposed rainfall signals using dynamic harmonic regressions: a Tropical Andean case study. Climate Dynamics, 52(7–8), 4643–4670. https://doi.org/10.1007/s00382-018-4400-3 Mora, D. E., & Willems, P. (2012). Decadal oscillations in rainfall and air temperature in the Paute River Basin-Southern Andes of Ecuador. Theoretical and Applied Climatology, 108(1–2), 267–282. https://doi.org/10.1007/s00704-011-0527-4 Tan, X., & Gan, T. Y. (2015). Nonstationary analysis of annual maximum streamflow of Canada. Journal of Climate, 28(5), 1788–1805. https://doi.org/10.1175/JCLI-D-14-00538.1 Vuille, M., Bradley, R. S., & Keimig, F. (2000). Climate Variability in the Andes of Ecuador and Its Relation to Tropical Pacific and Atlantic Sea Surface Temperature Anomalies. Journal of Climate, 13(14), 2520–2535. https://doi.org/https://doi.org/10.1175/1520-0442(2000)013<2520:CVITAO>2.0.CO;2

The revised manuscript is in the following link https://drive.google.com/drive/folders/1Yv5MROxxlbgPKX2pM5KG3Ji6v1MdtdiA?usp=sharing

[Figure]

Surucucho - Annual Peak Flow (APF)

[Figure]

Matadero - Annual Peak Flow (APF)

[Figure]

Surucucho - Annual Low Flow (ALF)

[Figure]

Matadero - Annual Low Flow (ALF)

**Fig. 1.**

(a)

R = 0.10
p-value = 0.59

R = 0.32
p-value = 0.15

R = 0
p-value = 0.96

APF (m³s⁻¹)

TNI (-)

R = 0.40
p-value = 0.03

R = 0.32
p-value = 0.29

R = 0.32
p-value = 0.21

APF (m³s⁻¹)

ENSO 1.2 (-)

(b)

R = 0.30
p-value = 0.11

R = 0.46
p-value = 0.03

R = 0
p-value = 0.94

APF (m³s⁻¹)

TNI (-)

R = 0.39
p-value = 0.03

R = 0.32
p-value = 0.18

R = 0.20
p-value = 0.57

APF (m³s⁻¹)

CAR (-)

**Fig. 2.**

Interactive
comment

[Figure]

Fig. 3.